# Role of CIV NS1 Protein in Innate Immunity and Viral Replication

**DOI:** 10.3390/ijms241210056

**Published:** 2023-06-13

**Authors:** Cheng Fu, Wenhui Zhu, Nan Cao, Wenjun Liu, Zhier Lu, Ziyuan Wong, Kaiting Guan, Chunyan Hu, Baoting Han, Sen Zeng, Shuangqi Fan

**Affiliations:** 1College of Animal Science & Technology, Zhongkai University of Agriculture and Engineering, Guangzhou 510225, China; 2College of Veterinary Medicine, South China Agricultural University, No. 483, Wushan Road, Tianhe District, Guangzhou 510000, China

**Keywords:** CIV, NS1 protein, MDA5, innate immune, viral replication

## Abstract

The innate immune pathway serves as the first line of defense against viral infections and plays a crucial role in the host’s immune response in clearing viruses. Prior research has indicated that the influenza A virus has developed various strategies to avoid host immune responses. Nevertheless, the role of the NS1 protein of the canine influenza virus (CIV) in the innate immune pathway remains unclear. In this study, eukaryotic plasmids of NS1, NP, PA, PB1, and PB2 were constructed, and it was found that these proteins interact with melanoma differentiation-associated gene 5 (MDA5) and antagonize the activation of IFN-β promoters by MDA5. We selected the NS1 protein for further study and found that NS1 does not affect the interaction between the viral ribonucleoprotein (RNP) subunit and MDA5, but that it downregulates the expression of the laboratory of genetics and physiology 2 (LGP2) and retinoic acid-inducible gene-I (RIG-I) receptors in the RIG-I pathway. Additionally, NS1 was found to inhibit the expression of several antiviral proteins and cytokines, including MX dynamin like GTPase 1 (MX1), 2′-5′oligoadenylate synthetase (OAS), Signal Transducers and Activators of Transcription (STAT1), tripartite motif 25 (TRIM25), interleukin-2 (IL-2), IFN, IL-8, and IL-1β. To further investigate the role of NS1, a recombinant H3N2 virus strain (rH3N2) and an NS1-null virus (rH3N2ΔNS1) were rescued using reverse-genetic technology. The rH3N2ΔNS1 virus exhibited lower viral titers compared to rH3N2, but had a stronger activation effect on the receptors LGP2 and RIG-I. Furthermore, when compared to rH3N2, rH3N2ΔNS1 exhibited a more pronounced activation of antiviral proteins such as MX1, OAS, STAT1, and TRIM25, as well as antiviral cytokines such as IL-6, IFN-β, and IL-1β. These findings suggest a new mechanism by which NS1, a nonstructural protein of CIV, facilitates innate immune signaling and provides new avenues for the development of antiviral strategies.

## 1. Introduction

Influenza A virus (IAV) is characterized by rapid transmission and a high mutation rate that poses a serious threat to both human and animal health [1]. Canine influenza virus (CIV) is an RNA virus that belongs to the Orthomyxoviridae family and causes respiratory symptoms, including coughing, runny nose, sneezing, and fever [2]. CIV was first identified in the United States and the main subtypes currently circulating are H3N8 and H3N2 [3,4]. Although both H3N8 and H3N2 are low-pathogenic viruses, coinfection of CIV with other respiratory diseases can cause more severe respiratory disease in dogs [5]. Furthermore, the emergence of different subtypes of CIV promotes recombination and evolution. For instance, the H1N1 human influenza virus and H3N2 CIV recombined to form the novel H3N1 CIV [6]. There are also differences in pathogenicity and epidemiology between the recombinant CIV and the original subtype of CIV [7].

The RNA genome of IAV comprises eight negative-sense RNA fragments encoding the viral proteins PB1, PB2, PA, NP, HA, NA, M1, and NS [8]. The NS gene of the influenza virus mainly encodes the nonstructural protein NS1 and the nuclear export protein NS2 [9]. NS1 is one of the earliest proteins expressed during viral replication and is an important multifunctional virulence factor of IAV [10,11]. NS1 protein affects IAV replication through several pathways. NS1, through its RNA-binding capacity and interacting proteins, promotes the splicing of M mRNA which encodes a variety of IAV proteins [12]. Moreover, NS1 promotes the nuclear export of viral late gene mRNAs by acting as an adaptation molecule between viral mRNAs and nuclear export factor 1 (NXF1) [13]. NS1 can also specifically enhance translation of viral mRNAs via interaction with the 5′UTR of the virus, the host nuclear poly (A) binding protein (PABP1), and eukaryotic initiation factor-4G (EIF4G) [14]. A study also found that NS1 interacts with the Staufen protein to enhance viral protein synthesis [15]. 

The NS1 protein is also involved in antagonizing host immune responses to IAV. The binding of NS1 to cleavage and polyadenylation specificity factor 4 (CPSF4) prevents the processing and maturation of host pre-mRNA and nuclear export. This causes a large accumulation of host pre-mRNA in the nucleus and inhibits the expression of IFN and other host proteins with antiviral effects [16]. Studies have found that NS1 interacts with the coiled-coil domain 2 of mixed-lineage kinase domain-like protein (MLKL) to increase MLKL oligomerization and membrane translocation [17]. Meanwhile, the interaction between MLKL and NS1 can enhance MLKL-mediated NOD-like receptor family pyrin domain-containing 3 (NLRP3) inflammasome activation which leads to increased IL-1β secretion [18]. However, the role and mechanism of NS1 in escaping host immune responses to CIV is still unclear.

The RIG-I-like receptors (RLRs) family of pattern-recognition receptors (PRRs) comprises a group of cytosolic RNA helicase proteins that can identify viral RNA as nonself via binding to pathogen-associated molecular pattern (PAMP) motifs within RNA ligands. [19]. RIG-I and MDA5 physically recognize IAV RNA, which causes several conformation changes, resulting in the recruitment of tripartite motif protein 25 (TRIM25) and other E3 ubiquitin ligases and ultimately activating key transcription factors nuclear factor kappa B (NF-κB) and interferon regulatory factor (IRF). NF-κB and IRF are responsible for producing IFN-I [20].

Numerous studies have shown that the influenza NS1 protein can escape host immunity through the RIG-I signaling pathway. A study demonstrated that pigeon RIG-I and the CARD region of MDA5 have a strong antiviral effect against AIV H9N2 in chicken DF-1 cells [21]. NSI directly interacts with the CARD region of RIG1 and the coiled-coil domain of TRIM25 to prevent ubiquitination of RIG-I [22]. Our previous study found that overexpression of the CARD region of MDA5 could inhibit CIV replication by activating the NF-κB and IRF3 promoters, which promotes cytokine release and the expression of antiviral proteins [23]. Another study also showed that the NS1 protein plays an important role in the anti-influenza activity of MDA5. NS1 from H5N1 HPAIV inhibited the duck MDA5-mediated signaling pathway in vitro [24]. Moreover, in the absence of functional NS1 antagonists of IAV, MDA5 is a significant contributor to cellular defense against IAV [25]. NS1 interacts directly with the RIG-I CARD domain, thus inhibiting RIG-I activation in a strain-specific manner. NS1 also interacts with various key proteins involved in the RIG-I pathway, such as the ubiquitin ligases TRIM25 and Riplet, and, thus, prevents RIG-I ubiquitination, oligomerization, and activation. The interaction of PB1-F2 with MAVS and IRF3 has also been correlated with reduced levels of IFN-β.

In this study, we constructed an NS1 eukaryotic expression plasmid and found that NS1 affects MDA5 function but not the interaction between NP, PA, PB1, PB2, and MDA5. Additionally, NS1 inhibits the production of antiviral proteins and cytokines downstream of RIG-I. Importantly, compared with rH3N2, rH3N2ΔNS1 exhibited higher viral titers and a stronger activation effect on the receptors LGP2 and RIG-I, the antiviral proteins MX1, OAS, STAT1, and TRIM25, and the cytokines IL-6, IFN-β, and IL-1β. Overall, our findings reveal the role of CIV NS1 protein in CIV replication and innate immunity.

## 2. Results

### 2.1. CIV Proteins Interact with MDA5

To explore the interaction between MDA5 and CIV proteins, bimolecular fluorescence complementation technology was used. Two complementary fragments of green fluorescent protein were connected to the MDA5 gene and the NS1 and vRNP (PB1/PB2/NP and PA) gene. When coexpressed in cells, if MDA5 and the candidate protein interact, the cells would emit fluorescence detectable by laser confocal microscopy. In 293T cells and MDCK cells, the NS1/PB1/PB2/NP and PA protein of the CIV binds to the MDA5 protein and produces green fluorescence, suggesting that MDA5 interacts with CIV proteins (Figure 1A). To further verify the association between RNP subunits and MDA5, laser confocal microscopy was used to reveal the colocalization of different RNP subunits and MDA5 in 293T cells (Figure 1B). The coexpression of PB1, PA, NP, and MDA5 mainly produces overlapping fluorescent signals in the cytoplasm, and PB2 mainly produces overlapping fluorescent signals in the nucleus. To verify that CIV proteins and MDA5 interact in vivo, we used a CIV-NP/NS1 antibody and an MDA5 antibody in CIV-infected or noninfected 293T cells; confocal images showed that NP, NS1, and MDA5 colocalized in the cytoplasm (Figure 1C). Finally, co-IP experiments were performed with 293T cells to confirm the interaction between MDA5 and NS1 or NA by transiently coexpressing p3xFLAG-NS1/vRNP with MYC-MDA5. After co-IP with anti-MYC, p3xFLAG-NS1, p3xFLAG-PA, p3xFLAG-PB1, p3xFLAG-PB2, and p3xFLAG-NP were found to form a complex with MYC-MDA5 but not with PCAGG-MYC (Figure 1D). 

### 2.2. NS1 Does Not Affect the Interaction between the RNP Subunits and MDA5

NS1 may compete with RNP subunits to bind MDA5, or it may inhibit the binding between RNP and MDA5. To solve this problem, NGFP-tagged NP, PA, PB1, PB2, and CGFP-tagged MDA5 were cotransfected with Flag-tagged NS1 or empty vectors into 293T cells. Interestingly, NS1 expression did not interfere with the binding of RNP subunits to MDA5 (Figure 2A above). Previous studies have shown that NS1 can selectively increase the synthesis of viral mRNA, resulting in the preferential synthesis of viral proteins. To test whether the BiFC signal was the result of NS1 stimulating RNP subunit synthesis, Western blot analysis was performed on cells transfected with the plasmid encoding the RNP subunit in the presence and absence of NS1, respectively (Figure 2A below). In addition, we transfected the MYC-NS1 or MYC-CMV plasmid into 293T cells infected with CIV (MOI = 0.1), and detected the relative mRNA expression of PA, PB1, PB2, and NP, and found that overexpression of NS1 had no significant effect on the expression of the above gene mRNA (Figure 2B). The data clearly showed that NS1 coexpression does not negatively affect the interaction of MDA5 and RNP subunits. 

### 2.3. Effect of CIV Protein on IFN-β Promoter Activated by MDA5

MDA 5 promotes IFN-β production and induces a wide range of nonspecific antiviral infections, while also modulating immune responses. Therefore, we tested the luciferase reporter gene driven by the IFN-β promoter to determine whether the CIV proteins affect the production of IFN triggered by MDA5. The MYC-tagged MDA5 was cotransfected with NS1-, PB1-, PB2-, PA-, and NP-expressing plasmids in 293T cells for 24 h, and then stimulated with poly I:C to induce IFN synthesis. Compared with untreated cells, 293T cells treated with MDA5 and poly I:C resulted in a significant increase in INF-β promoter induction. However, only NS1, PB1, and PA had inhibitory effects on MDA5-induced IFN-β promoter activity, among which NS1 had the most obvious inhibitory effect. NP and PB2 did not show statistically significant inhibition (Figure 3A). At the same time, the expressions of MDA5 and CIV proteins were detected by Western blot, and it was found that CIV proteins inhibited the expression of the MDA5 protein, except PB1 and PA (Figure 3B,C). To this end, we conducted an in-depth study on the role of the NS1 protein in CIV replication. 

### 2.4. Overexpression of NS1 Protein Antagonizes the Activation of IRF3 and IFN-β Promoters by the CARD Region of MDA5

MDCK cells were transiently cotransfected with either p3xFLAG-NS1 and MYC-MDA5-CARD or p3xFLAG and MYC-MDA5-CARD, along with reporter genes (PGL3-IFN-β, pGL3-IRF3, and PGL3-NF-κB) and the reference gene PRL-TK. After 36 h, reporter gene fluorescence was measured using a dual-luciferase detection system. The results showed the activation effect of MYC-MDA5-CARD on the IFN-β promoter was antagonized by the influenza virus NS1 protein (Figure 4A). Moreover, the NS1 protein of H3N2 significantly antagonized the activation of the IRF3 promoter by MYC-MDA5-CARD (*p* < 0.001) (Figure 4B). However, the antagonistic effect of the NS1 protein on the NF-κB promoter was not obvious (Figure 4C). Meanwhile, Western blotting to detect the expression of IRF3 revealed that NS1 could reduce the activation effect of MDA5-CARD on IRF3, IFN, IFN-β, and P65 (Figure 4D). Finally, we examined the effect of MDA5-CARD on IFNβ and IRF3 mRNA expression in cells transfected with Flag-CMV or Flag-NS1 and found that the NS1 protein of H3N2 significantly antagonized the promotion of IFNβ and IRF3 mRNA by MYC-MDA5-CARD (*p* < 0.05) (Figure 4E). The results indicated that the NS1 protein of H3N2 CIV could significantly antagonize the activation of the IRF3 and IFN-β promoters by the CARD region of MDA5, but not the NF-κB promoter.

### 2.5. NS1 Protein Inhibited the Expression of the RIG-I Pathway and Its Downstream ISGs and Cytokines

MYC-MDA5-CARD was transiently cotransfected with either p3xFLAG-NS1 or p3xFLAG in MDCK cells. After 36 h, RT-qPCR was used to detect the expression of the RIG-I pathway and its downstream ISG genes. The results showed that the NS1 gene of CIV could effectively downregulate the expression of LGP2 and RIG-I receptors in the RIG-I pathway (Figure 5A). Further studies showed that NS1-overexpressing CIV antagonized ISG genes such as MX1, OAS, STAT1, and TRIM25, especially MX1 (*p* < 0.001), but not IPS1 (Figure 5B). At the same time, NS1-overexpressing CIV significantly inhibited the expression of cytokines IL-2, IFN-β, IL-8, and IL-1β (Figure 5C). We selected RIG-I, STAT1, and TRIM25 proteins and detected their expression by Western blotting. NS1 overexpression inhibited the activation effect of the MDA5-CARD region on RIG-I, STAT1, and TRIM25. In conclusion, the NS1 protein of influenza virus can antagonize the RIG-I pathway and its downstream proteins that are regulated by the CARD region of MDA5.

### 2.6. Recombinant NS1-Null Virus Strains Promoted Activation of the IFN-β and IRF3 Promoters

The recombinant H3N2 virus strain (rH3N2) and NS1-null virus (rH3N2ΔNS1) were rescued by reverse-genetic technology. MDCK cells were transfected with IFN-β and IRF3 luciferase plasmids, and the recombinant viruses rH3N2 and rH3N2ΔNS1 were infected at different doses (MOI = 0.01, MOI = 0.05, MOI = 0.1). The fluorescence value was detected by a dual-luciferase assay 36 h later. The results showed that the recombinant NS1-null virus had a stronger activation effect on the promoters of IFN-β and IRF3, and the effect was strongest at 0.01 MOI (Figure 6). 

### 2.7. Deletion of NS1 Protein Inhibited H3N2 Replication

MDCK cells were transfected with MYC-MDA5-CARD or MYC-CMV and infected with the recombinant viruses rH3N2 and rH3N2ΔNS1 at an MOI of 0.1. CIV titers were detected by TCID50 in MDCK cells as calculated by the Reed–Muench method at different times. The results showed that the NS1 protein deletion and MYC-MDA5-CARD inhibited CIV replication (Figure 7A). To demonstrate the growth deficiency of the rH3N2 and rH3N2ΔNS1 viruses, growth kinetics in A549 cells with and without poly I:C pretreatment were measured. Poly I:C treatment decreased the titers of the rH3N2 and rH3N2ΔNS1 viruses at 48 h (Figure 7B).

### 2.8. Deletion of NS1 Protein Inhibited Innate Immunity 

After MDCK cells were infected with different doses of the recombinant viruses rH3N2 and rH3N2ΔNS1, RNA was extracted from the cells and the expression of RIG-I pathway genes and their downstream ISG genes was detected by qPCR. In rH3N2- and rH3N2ΔNS1-infected cells, the NS1-null virus significantly upregulated the RIG-I pathway pattern-recognition receptors MDA5, RIG-I, and LGP2. The upregulated effect was most obvious at 0.01 MOI. It also significantly upregulated IFN-β and the downstream genes of ISGs, such as STAT1, IPS1, MX1, OAS, and TRIM25. Again, the upregulation was most obvious at MOI = 0.01. It also upregulated the cytokines, with the upregulated effect being the most obvious at MOI = 0.01 (Figure 8).

## 3. Discussion

Due to the close human–animal interface, the study of the mechanism and pathogenicity of CIV is important [26]. The NS1 protein is an important virulence factor of IAV that can affect the antiviral activity in vivo through its antagonistic effect on interferon signaling [10]. NS1 also inhibits interferon expression by binding to double-stranded RNA (dsRNA), and studies have shown that NS1 inhibits interferon synthesis by inhibiting activator protein-1 (AP-1) transcription [27]. Moreover, using reverse-genetic techniques, NS1-deficient H3N8 and H3N2 CIV strains have been constructed and used in live-attenuated influenza vaccines for the control and prevention of influenza [28,29,30,31].

The innate immune pathway is the host’s first line of defense against viral infection and plays a key role in viral clearance [32,33]. For example, the cGAS-cGAMP-STING pathway exhibits a wide range of antiviral activities [34]. During the invasion of most DNA viruses (e.g., herpesviruses and retroviruses), the transcription of IFN-I is initiated in the host via activation of the cGAS-cGAMP-STING pathway, inducing an effective antiviral state in the host [35,36,37]. However, DNA viruses evade or actively inhibit the activation of cGAS or STING [38], and a few RNA viruses have also been reported to inhibit the cGAS-cGAMP-STING pathway [39,40]. For example, the evolutionary mechanism of DENV invasion is mediated through inhibition of IFN-I induction by antagonizing the cGAS-cGAMP-STING signaling pathway [41]. A large number of studies have shown that during IAV infection, the production of cytokines such as IFN-I can be induced in the host through the RIG-I-receptor-mediated signaling pathway and play an antiviral role [42]. The current study identified three major members of the RIG-I-like receptor family, namely, RIG-I, MDA5, and LGP2, all of which have RNA helicase domains containing special DEX/DH frames that bind to RNA and act as ATPases to promote RNA conformational changes and activate downstream signal transduction [43]. The transcription and activation of RIG-I are important for IAV replication [44]. A study reported the biological effects of NS1 binding to RIG-I, which directly silenced alarms that activated cellular innate immunity against infection [22]. In this study, we found that NS1 and MDA5 interacted in 293T cells (Figure 1), but we did not verify whether the interaction between MDA5 and NS1 is mediated by nucleic acid, and the RNase enzyme need be added in the lysis buffer during co-IP for the further experiment. Furthermore, we found that NS1 does not affect the interaction between the RNP subunits and MDA5 (Figure 2).

RIG-I-like receptors recognize viral components and induce the production of IFN-I, subsequently activating MAVS recruitment to the downstream mediators TNF receptor-associated factor 3 (TRAF3), TANK-binding kinase 1 (TBK1), and I-kappaB kinase (IKK-I), leading to phosphorylation-mediated IRF3 activation [45]. TRAF3, TBK1, and IKK-I are common molecules produced by toll-like receptor (TLR) pathway activation via IFN-I. MAVS recruits TRAF3 and undergoes K63-linked ubiquitination and then recruits the activating kinases TBK1 and IKK-I to phosphorylate IRF3/IRF7, which are translocated to the nucleus after their activation to initiate IFN-I gene transcription [46,47]. TRIM25, another molecule that regulates the RIG-I-like receptor (RLR) pathway, mainly enhances the binding of RIG-I to MAVS, thereby promoting the activation of the RIG-I antiviral signaling pathway. Our previous study showed that the canine MDA5 gene plays an important role in CIV replication [48]. In this study, we found that only NS1, PB1, and PA had inhibitory effects on MDA5-induced IFN-β promoter activity, among which NS1 had the most pronounced inhibitory effect (Figure 3). Moreover, overexpression of NS1 inhibited LGP2 and RIG-I receptor expression (Figure 5A). Compared with rH3N2, the NS1-null virus (rH3N2ΔNS1) significantly upregulated MDA5, RIG-I, and LGP2 in the RIG-I pathway (Figure 8), indicating that CIV NS1 has an inhibitory effect on the RIG-I signaling pathway.

After overexpressing the CARD of MDA5, we found that the MDA5-mediated signaling pathway transmits signals downstream by activating the IRF3 promoter. Specific functional silencing of MDA5 in vivo decreased cytokine production, weakened antiviral activity, and enhanced CIV replication, indicating that MDA5 inhibits CIV replication primarily through its CARD. In addition, the CARD of MDA5 significantly activated the expression of two other pattern-recognition receptors, LGP2 and RIG-I, and activated the RIG-I-like pathway and its downstream ISG pathway, promoting the expression of antiviral proteins and the release of cytokines, thus inhibiting virus replication [23]. In this study, we found that NS1 significantly inhibited the activation of IFN-β and IRF3 via the MDA5 pathway but had no significant effect on the activity of the NF-κB signaling pathway (Figure 4). Furthermore, we detected the expression of antiviral proteins such as MX in the downstream RIG-I pathway and found that overexpression of NS1 significantly inhibited the mRNA expression of MX1, OAS, STAT1, and TRIM25, especially MX1 (Figure 6B). Moreover, compared with rH3N2, rH3N2ΔNS1 significantly enhanced the mRNA expression of MX1 and TRIM25 (Figure 8), indicating that CIV NS1 exerted significant inhibitory effects on antiviral proteins in the downstream RIG-I pathway.

Eukaryotic expression of the NS1 gene inhibits interferon activity regulated by canine MDA5, similar to the effect of the NS1 gene on MDA5 in poultry [49]. In addition, NS1 can inhibit inflammatory and antiviral signaling pathways regulated by MDA5 in dogs, consistent with previous studies in other mammals [50]. Several studies have shown that NS1-null influenza viruses elicit a major IFN response during cell infection [51,52]. In our study, a reverse-genetics approach was used to rescue the rH3N2 and rH3N2ΔNS1 viruses, and compared with rH3N2, rH3N2ΔNS1 had a lower degree of viral shedding (Figure 7) but significantly induced the mRNA expression of IFN-β and inflammatory cytokines (Figure 8).

In summary, our study provides the first evidence that the CIV NS1 protein antagonizes the function of MDA5 and inhibits the expression of antiviral proteins and cytokines. Our findings further shed light on the role of NS1 in CIV replication and innate immunity, suggesting a promising avenue for the development of antiviral strategies.

## 4. Materials and Methods

### 4.1. Viruses and Cells

The H3N2 CIV strain (A/canine/Guangdong/02/2011, C/GD/02) used in the study was isolated in 2011 from a pet dog with typical symptoms. Viruses were propagated in the allantoic cavity of 9-day-old eggs from specific pathogen-free chickens by incubation at 37 °C for 72 h. Viral titers are expressed as values of 50% tissue culture infective dose (TCID50)/0.1 mL in Madin–Darby canine kidney (MDCK) cells as calculated by the Reed–Muench method. MDCK cells and 293T cells were grown in Dulbecco’s modified Eagle’s medium (Gibco, Grand Island, NY, USA) supplemented with 10% fetal bovine serum (Thermo Fisher Scientific, Waltham, MA, USA, 10099). Cells were cultured at 37 °C in a 5% CO_2_ incubator.

### 4.2. Reagents and Antibodies

The chemical reagent poly I:C used in this study was purchased from Sigma–Aldrich (CAS Number: P1530). The primary antibodies used in this study were as follows: Mouse monoclonal anti-Tubulin (Beyotime, Haimen, China, AT819), rabbit monoclonal anti-Flag (Cell Signaling Technology, Danforth, MA, USA, 14793), mouse monoclonal anti-Myc (Beyotime, AF2864), rabbit polyclonal anti-stat-1 α/β (Beyotime, AF0288), mouse monoclonal anti-IRF3 (Beyotime, AG2321), rabbit monoclonal anti-phospho-IRF3 (Ser386) (Beyotime, AF1594), rabbit monoclonal anti-RIG-I (D33H10) (Cell Signaling Technology, 4200), and rabbit monoclonal TRIM25 (D9T7G) (Cell Signaling Technology, 13773). The secondary antibodies used for immunoblot analysis were HRP-conjugated goat anti-mouse IgG (Bioworld Technology, St Louis Park, MN, USA, BS12478) and HRP-conjugated goat anti-rabbit IgG (Bioworld Technology, BS13278).

### 4.3. Quantitative Real-Time RT-PCR (qPCR)

For targeted gene expression analysis, total RNA was extracted from cells, and complementary DNA (cDNA) was synthesized using a PrimeScript RT Master Mix reverse-transcription kit (Takara). Real-time qPCR was carried out using a LightCycler 480 instrument (Roche). Relative mRNA expression was calculated from triplicate samples using the 2^−ΔΔCt^ method and normalized to the expression of the housekeeping gene GAPDH. The primers used are listed in Table 1.

### 4.4. Construction of Plasmids

The sequences of NS1, PA, PB1, PB2, NP, MDA5, and MDA5 with deletion of the N-terminal CARD were inserted into p3xFLAG-CMV and MYC-CMV as appropriate. The constructed plasmids were named p3xFLAG-NS1, p3xFLAG-PA, p3xFLAG-PB1, p3xFLAG-PB2, p3xFLAG-NP, MYC-MDA5, and MYC-MDA5-CARD.

### 4.5. Transfection

MDCK cells or 293T cells grown to 60% confluence in 6-well cell culture plates were transfected with plasmids using Lipofectamine^®^ 3000 reagent (L3000015, Thermo Fisher, Grand Island, NY, USA) according to the manufacturer’s instructions. Briefly, 1 μg of plasmids and 2 μL of P3000 were diluted in 50 μL of serum-free Opti-MEM. Furthermore, 3 μL of Lip3000 was diluted in 50 μL of serum-free Opti-MEM. The dilutions were mixed thoroughly and incubated at 25 °C for 15 min. The mixtures were then pipetted into the medium, and cells were further cultured at 37 °C for 24 h. 

### 4.6. Luciferase Assay

MDCK cells or 293T cells were cultured in 24-well plates and grown to approximately 70% confluence at 37 °C in a 5% CO2 incubator. The canine IFN-β promoter, IRF-3 response element, and NF-κB response element constructs were generated based on previous studies. 293T cells were transiently transfected with various expression plasmids (p3xFLAG-NS1, PA, PB1, PB2, NP, and MYC-MDA5, or empty control plasmid) together with luciferase-expressing plasmids pGL3-IFN-β and pRL-TK using Lipofectamine 3000,and MDCK cells were transiently transfected with various expression plasmids (p3xFLAG-NS1, MYC-MDA5-CARD, or empty control plasmid) together with luciferase-expressing plasmids (pGL3-IFN-β, pGL3-IRF3, and pGL3-NF-κb) and pRL-TK using Lipofectamine 3000. According to the manufacturer’s instructions, cells were lysed and collected for measurement of luciferase activities using the Dual-Luciferase Assay Kit (Promega, Madison, WI, USA) 36 h post-transfection. All luciferase reporter assays were repeated three times.

### 4.7. Rescue of rH3N2 and rH3N2ΔNS1 Viruses

The rH3N2 and rH3N2ΔNS1 viruses were rescued as previously described [53,54]. 293T cells were uniformly seeded in 6-well plates and transfected with Lipofectamine 3000 reagent at 80% confluence. Briefly, two 1.5 mL EP tubes were selected. Then, 125 μL of Opti-MEM and 1 μg of PB1, PB2, PA, and NP were added into tube A, mixed, and incubated at room temperature for 5 min to generate ribonucleoprotein complexes; then, 1 μg each of the HA, NA, M1, and NS plasmids was added (rH3N2 was supplemented with the NS plasmid, and rH3N2ΔNS1 was supplemented with the NS1 plasmid without an open reading frame), followed by 10 μL of p3000. Then, 125 μL of Opti-MEM and 7.5 μL of lip3000 were added to tube B. After incubation, all the solution in tube B was added to the 6-well plate, and the 6-well plate was gently shaken to evenly distribute the solution. Six hours later, the medium in the 6-well plates was replaced with complete medium containing 5% FBS and 1% double antibiotic. The supernatant was collected after 48 h of culture, and 2 μL of TPCK trypsin was added and further incubated for 2 h. The cell supernatant and cell mixture were inoculated into 9-day-old embryos of SPF chickens. After 72 h of culture at 37 °C, allantoic fluid was collected, and hemagglutination activity was determined by an HA assay. 

### 4.8. Bimolecular Fluorescence Complementation (BiFC)

The NGFP-targeted NS1, PA, PB1, PB2, PA, and MDA5-CGFP plasmids were cotransfected into 293T cells, cultured in a 37 °C, 5% CO2 incubator for 24 h, and then incubated at 30 °C for 14 h. The samples were fixed with 4% paraformaldehyde at 4 °C for 30 min, permeabilized with 0.2% Triton X-100 permeabilization agent for 15 min, and then incubated sequentially with a mouse monoclonal anti-GFP antibody and a fluorescent anti-mouse secondary antibody. Finally, the samples were observed by laser confocal microscopy after DAPI staining.

### 4.9. Coimmunoprecipitation (co-IP) and Western Blotting

To confirm the interaction between MDA5 and CIV proteins, 293T cells were transfected with the 3xFlag-tagged NS1, PA, PB1, PB2, NP, and MYC-tagged MDA5 plasmids, harvested 48 h post-infection (pi), washed three times with cold PBS, and lysed with NP-40 buffer (Sigma–Aldrich, 127087–87-0) containing 1 mM phenylmethylsulfonyl fluoride (PMSF; Beyotime, ST506) for 1 h at 4 °C. Clarified extracts were precleared with protein A/G beads (Santa Cruz Biotechnology, sc-2003) plus an anti-Flag monoclonal antibody for 4 h, washed with NP-40 buffer, boiled in sample buffer, and subjected to sodium dodecyl sulfate—polyacrylamide gel electrophoresis (SDS-PAGE). This was followed by immunoblot analysis with anti-Flag and anti-MYC antibodies.

### 4.10. Confocal Immunofluorescence Microscopy

293T cells were grown in 35 mm petri dishes (NEST, GBD-35-20) with a glass bottom. When needed, the indicated plasmid DNA (Flag-tagged NS1, NP, PA, PB1, and PB2, and MYC-tagged MDA5) was transfected for the interaction between MDA5 and the proteins of CIV. Cells were washed with PBS and fixed with 4% paraformaldehyde (Sigma–Aldrich, St. Louis, MO, USA, P6148) for 30 min at room temperature; they were then permeabilized with 0.2% triton X-100 (Sigma–Aldrich, T8787) for 10 min. The cells were blocked in PBS containing 5% bovine serum albumin (BSA; Beyotime, ST023) for 30 min. Next, the cells were stained with the indicated primary antibody of rabbit polyclonal antibody (anti-Flag; 1:400) and a mouse monoclonal antibody (anti-MYC; 1:200) in PBS buffer at 37 °C, followed by a 1 h incubation in PBS containing goat anti-mouse and anti-rabbit secondary antibodies conjugated to FITC and TRITC at a dilution of 1:200. Wherever indicated, nuclei were stained with DAPI (Beyotime, C1002). The fluorescence signals were visualized with a TCS SP2 confocal fluorescence microscope (Leica TCS SP8).

### 4.11. Statistical Analysis

Statistical analysis was performed with unpaired Student’s *t*-tests and two-way ANOVA, as implemented in GraphPad Prism 5 software (mean ± SD; *n* = 3; * *p* < 0.05; ** *p* < 0.01; *** *p* < 0.001; **** *p* < 0.0001; ns *p* > 0.05).

## Figures and Tables

**Figure 1 ijms-24-10056-f001:**
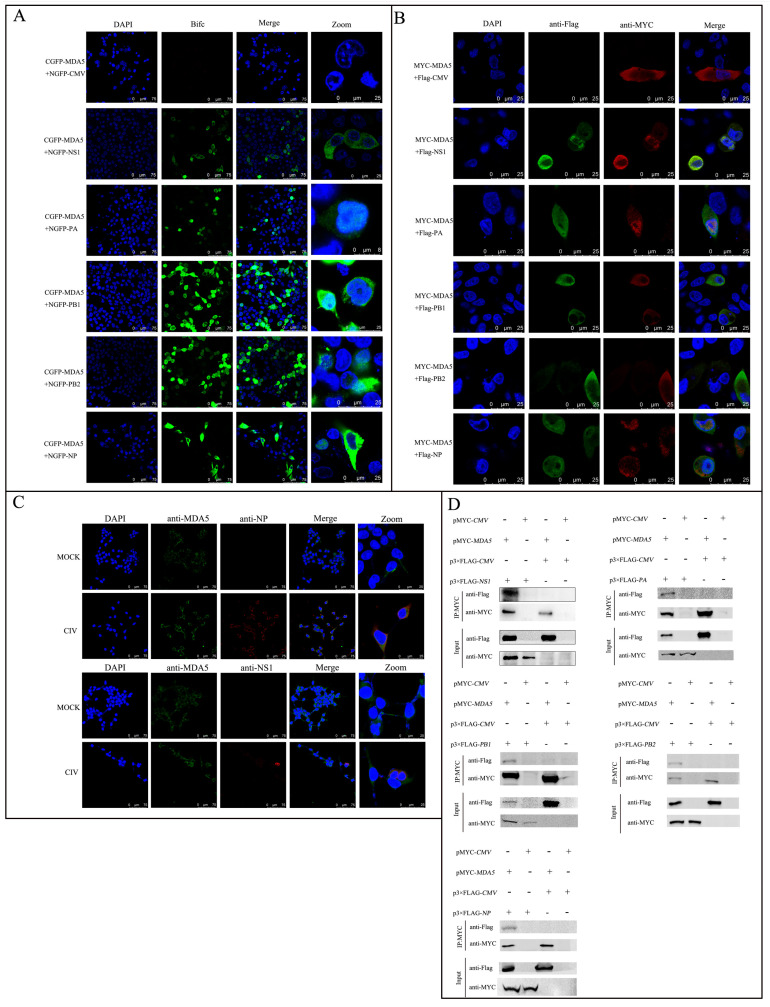
CIV proteins interact with MDA5. (**A**) Colocalization of NS1, NP, PA, PB1, and PB2 with MDA5. 293T cells were cotransfected with NGFP-tagged NS1, NP, PA, PB1, and PB2 (2.5 µg/well) and CGFP-tagged MDA5 (2.5 µg/well) in a 14 mm confocal dish. Cells were fixed at 36 h post transfection and observed by laser confocal microscopy. The nucleus is indicated by DAPI (blue) staining in the merged image. (**B**) 293T cells were cotransfected with Flag-tagged NS1, NP, PA, PB1, PB2, and MYC-tagged MDA5. Cells were fixed at 48 h post-transfection and subjected to an indirect immunofluorescence assay to detect MYC-MDA5 (red) and Flag-tagged NS1, NP, PA, PB1, PB2 (green) with mouse anti-Flag and rabbit anti-MYC antibodies. The nucleus is indicated by DAPI (blue) staining in the merged image. (**C**) Colocalization of NS1, NP protein with MDA5. 293T cells were mock-infected or infected with CIV (MOI = 0.1). Cells were fixed at 48 h post-transfection and subjected to an indirect immunofluorescence assay to detect MDA5 (green) and NS1/NP (red) with mouse anti-NS1/NP and rabbit anti-MDA5 antibodies. The nucleus is indicated by DAPI (blue) staining in the merged image. (**D**) Coimmunoprecipitation (co-IP) analysis of Flag-tagged NS1, NP, PA, PB1, PB2, and MYC-tagged MDA5 by the anti-Flag monoclonal antibody (mAb) or by the anti-MYC mAb. 293T cells were cotransfected with the indicated plasmids (+) or empty vectors (−) for 48 h. The transfected cells were lysed and incubated with a mouse anti-MYC mAb, followed by incubation with protein G-agarose for 6 h at 4 °C. The immunoprecipitate was analyzed by Western blotting using anti-Flag and anti-MYC antibodies.

**Figure 2 ijms-24-10056-f002:**
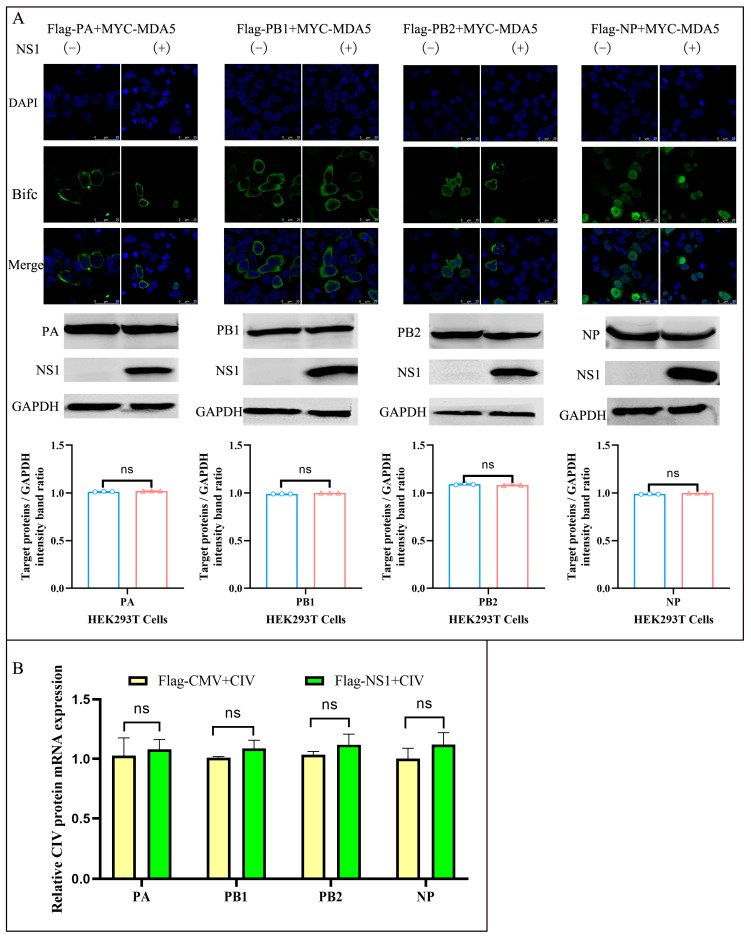
NS1 does not affect the interaction between the RNP subunit and MDA5. (**A**) NP, PA, PB1, and PB2 interact with MDA5 in 293T cells with an NS1 plasmid or empty plasmid. 293T cells were cotransfected with NGFP-tagged NP, PA, PB1, and PB2 (2.5 µg/well) and CGFP-tagged MDA5 (2.5 µg/well), followed by transfection with a Flag-tagged NS1 or empty plasmid in a 14 mm confocal dish. Cells were fixed at 36 h post transfection and observed by laser confocal microscopy. The nucleus is indicated by DAPI (blue) staining in the merged image (Above). After transfection with the targeted plasmid 24 hpi, the NS1 or NP, PA, PB1, or PB2 protein expression in 293T cells was assessed using a Western blot assay as described in Materials and Methods; the level of protein was quantified using Image-Pro Plus 6.0 software, the circles, small triangles represent the quantification is repeated three times, and the red and blue columns represent the transfected and untransfected NS1 plasmids respectively. Error bars indicate the mean (±SD) of 3 independent experiments; ns *p* > 0.05 (one-way ANOVA (below)). (**B**) 293T cells were transduced with MYC-NS1 or MYC-CMV for 24 h, followed by infection with CIV at an MOI of 0.1. The NP, PA, PB1, and PB2 mRNA expression in 293T cells was assessed using a real-time RT-PCR assay as described in Materials and Methods.

**Figure 3 ijms-24-10056-f003:**
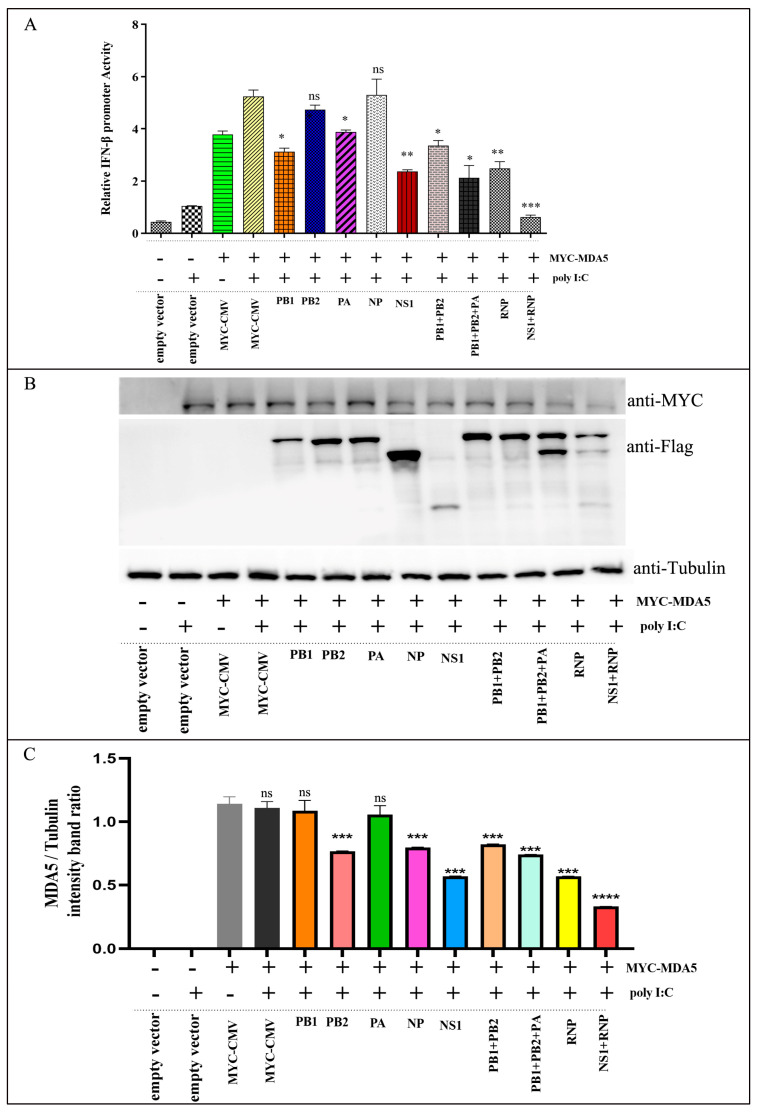
CIV proteins inhibit the activation of IFN-β promoters by the MDA. (**A**) p3xFLAG, p3xFLAG-NS1, p3xFLAG-NP, p3xFLAG-PB1, p3xFLAG-PB2, p3xFLAG-PA, and Flag-MDA5-CARD (500 ng/well) were transfected with pGL3-IFN-β (100 ng/well) and pRL-TK (50 ng/well) in 24-well plates for 24 h followed by stimulation with poly I:C. Next, cells were harvested, and the relative fluorescence intensities of IFN-β was measured 36 h later. (**B**,**C**) At the same time, cell samples were analyzed by immunoblotting with antibodies against MYC, Flag, and Tubulin (loading control). The data were analyzed by one-way ANOVA with post hoc correction for multiple comparisons with Fisher’s least significant difference (LSD) method. * *p* < 0.05; ** *p* < 0.01; *** *p* < 0.001; **** *p* < 0.0001; ns *p* > 0.05. Error bars indicate the mean (±SD) of 3 independent experiments.

**Figure 4 ijms-24-10056-f004:**
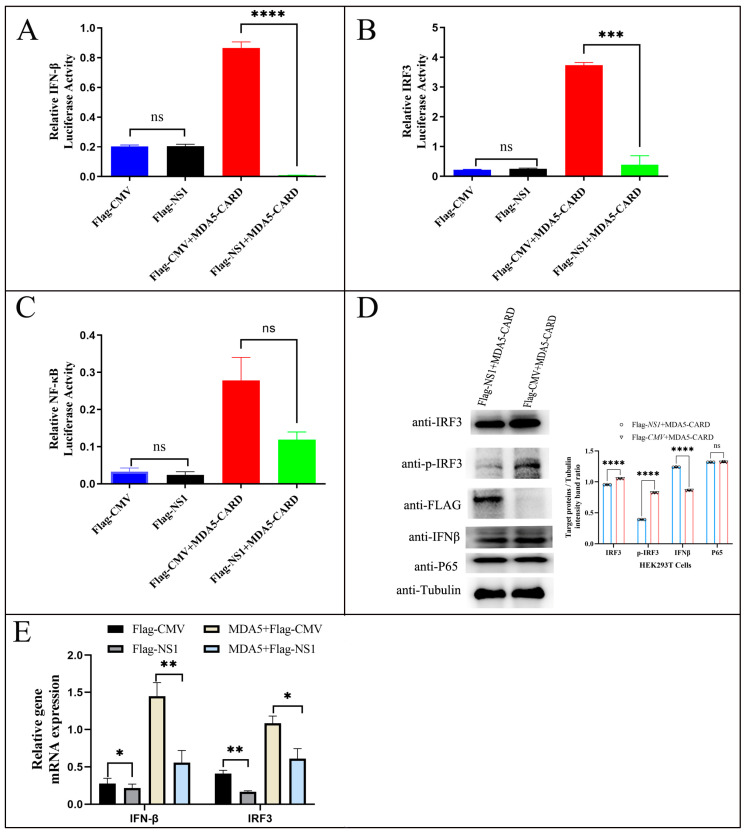
NS1 inhibits the activation of IFN-β and IRF3 promoters by the CARD of MDA. Effect of NS1 on IFN-β/IRF3 and NF-κB promoter activation by the CARD of MDA5. p3xFLAG-CMV, p3xFLAG-NS1, and MYC-MDA5-CARD (500 ng/well) were transfected with pGL3-IFN-β (100 ng/well) (**A**), pGL3-IRF3 (100 ng/well) (**B**), pGL3-NF-κB (100 ng/well) (**C**), and pRL-TK (50 ng/well) in 24-well plates. Next, cells were harvested, and the relative fluorescence intensities of IFN-β, IRF3, and NF-κB were measured 36 h later. The data were analyzed by one-way ANOVA with post hoc correction for multiple comparisons with Fisher’s least significant difference (LSD) method. * *p* < 0.05; ** *p* < 0.01; *** *p* < 0.001; **** *p* < 0.0001; ns *p* > 0.05. Error bars indicate the mean (±SD) of 3 independent experiments. (**D**) MYC-MDA5-CARD was transiently cotransfected with p3xFLAG-NS1 or p3xFLAG into 293T cells for 24 h, and cell samples were analyzed by immunoblotting with antibodies against IRF3, p-IRF3, IFN-β, P65, Flag, and Tubulin (loading control). (**E**) Effect of NS1 on IFN-β and IRF3 mRNA expression by MDA5. p3xFLAG-CMV, p3xFLAG-NS1, and MYC-MDA5 in 293T cells. Next, cells were harvested, and the relative mRNA expression of IFN-β and IRF3 was measured 24 h later using an RT-qPCR assay as described in Materials and Methods.

**Figure 5 ijms-24-10056-f005:**
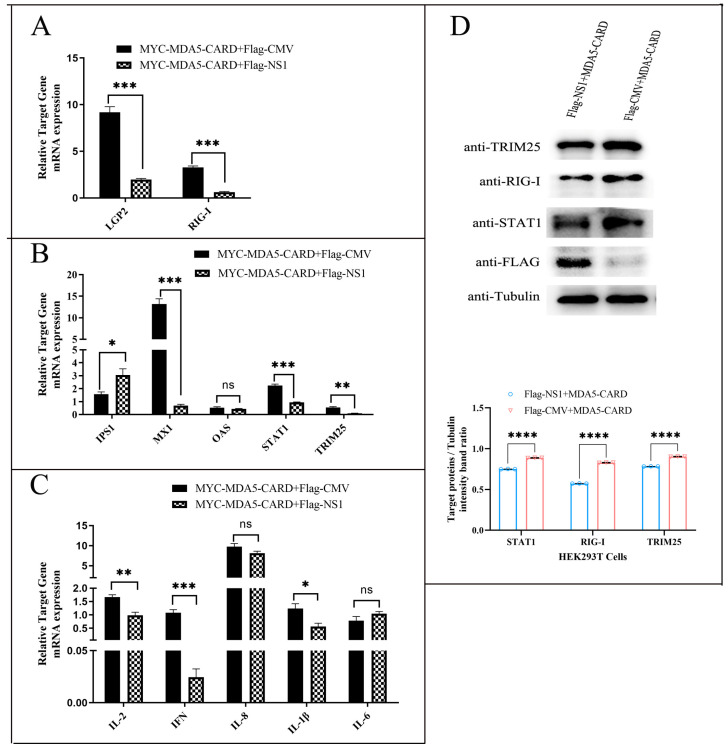
NS1 inhibits the MDA5-CARD-regulated pathway. MYC-MDA5-CARD was transiently cotransfected with p3xFLAG-NS1 or p3xFLAG (2.5 µg/well) into MDCK cells. The relative mRNA expression levels of (**A**) RLR (LGP2 and RIG-I), (**B**) antiviral molecules (IPS1, Mx, OAS, and STAT1), and (**C**) proinflammatory cytokines (IFN-β, IL-1β, IL-2, IL-6, and IL-8) in MDCK cells were determined by RT-qPCR after 36 h of transfection. The data were analyzed by one-way ANOVA with post hoc correction for multiple comparisons with Fisher’s least significant difference (LSD) method. * *p* < 0.05; ** *p* < 0.01; *** *p* < 0.001; **** *p* < 0.0001; ns *p* > 0.05. Error bars indicate the mean (±SD) of 3 independent experiments. (**D**) The protein expression of TRIM25, RIG-I, and STAT1 was analyzed by immunoblotting with the appropriate antibodies. NS1 expression was detected with an anti-Flag antibody, and Tubulin was used as a loading control.

**Figure 6 ijms-24-10056-f006:**
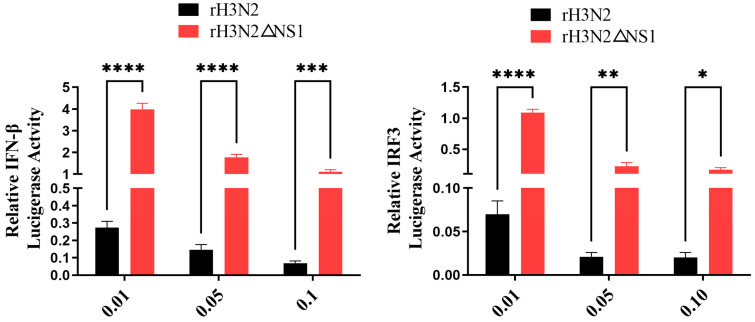
The NS1-null virus had a stronger activating effect on the promoters of IFN-β and IRF3. 293T cells were transfected with pGL3-IFN-β (100 ng/well), pGL3-IRF3 (100 ng/well), and pRL-TK (50 ng/well) in 24-well plates. Next, the cells were infected with different doses (MOI = 0.01, MOI = 0.05, MOI = 0.1) of the recombinant viruses rH3N2 and rH3N2ΔNS1. The relative fluorescence intensities of IFN-β and IRF3 were measured after 36 h. The data were analyzed by one-way ANOVA with post hoc correction for multiple comparisons with Fisher’s least significant difference (LSD) method. * *p* < 0.05; ** *p* < 0.01; *** *p* < 0.001; **** *p* < 0.0001. Error bars indicate the mean (±SD) of 3 independent experiments.

**Figure 7 ijms-24-10056-f007:**
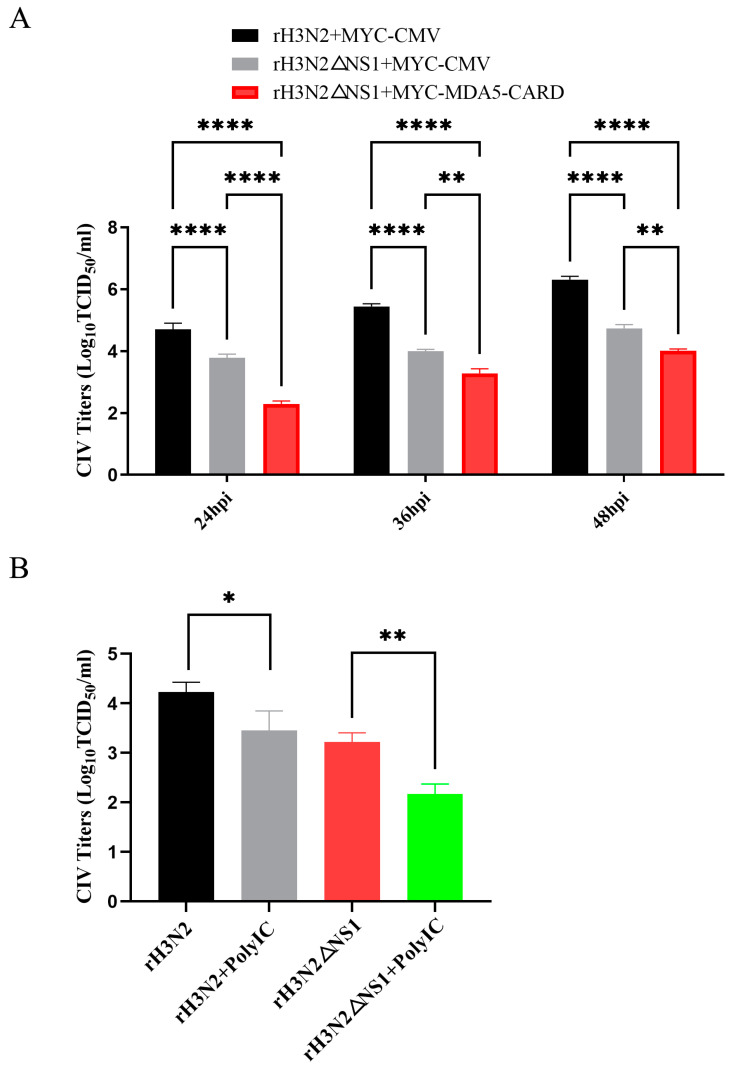
Deletion of the NS1 protein and the MDA5-CARD reduced the viral titer of H3N2. (**A**) MDCK cells were transiently transfected with 2.5 μg/well MYC-MDA5-CARD or MYC-MDA5 in 6-well plates. After 24 h, the MDCK cells were infected with 0.1 MOI of the recombinant viruses rH3N2 and rH3N2ΔNS, and supernatants were collected 24 h, 36 h, and 48 h after infection. The titer of the replicating virus was measured by a TCID50 assay. (**B**) A549 cells were pretreated with and without poly I:C (2.5 µg/well) in 6-well plates. After 24 h, the A549 cells were infected with 0.1 MOI of the recombinant viruses rH3N2 and rH3N2ΔNS, and supernatants were collected 24 h after infection. The data were analyzed by one-way ANOVA with post hoc correction for multiple comparisons with Fisher’s least significant difference (LSD) method. * *p* < 0.05; ** *p* < 0.01; **** *p* < 0.0001. Error bars indicate the mean (±SD) of 3 independent experiments.

**Figure 8 ijms-24-10056-f008:**
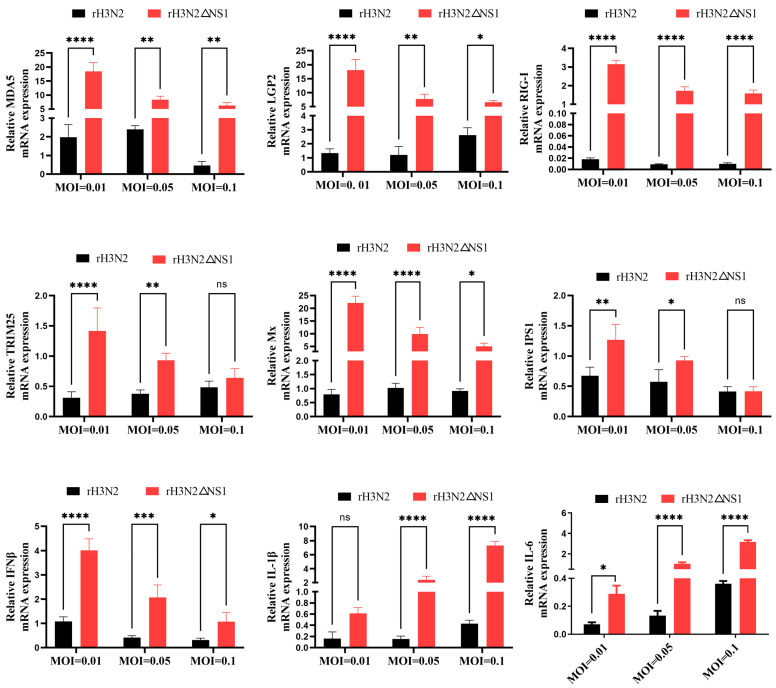
Activation of the MDA5 regulatory pathway by the rH3N2ΔNS1 virus. MDCK cells were infected with different doses (MOI = 0.01, MOI = 0.05, MOI = 0.1) of the recombinant viruses rH3N2 and rH3N2ΔNS1. The relative mRNA expression levels of RLR family members (MDA5, LGP2, and RIG-I), antiviral molecules (TRIM25, Mx, and IPS1), and proinflammatory cytokines (IFN-β, IL-1β, and IL-6) in MDCK cells were determined by RT-qPCR after 24 h of infection. The data were analyzed by one-way ANOVA with post hoc correction for multiple comparisons with Fisher’s least significant difference (LSD) method. * *p* < 0.05; ** *p* < 0.01; *** *p* < 0.001; **** *p* < 0.0001; ns *p* > 0.05. Error bars indicate the mean (±SD) of 3 independent experiments.

**Table 1 ijms-24-10056-t001:** Primers used for real-time PCR quantitation.

Gene	Sequence (5′~3′)
IL-1β	F: TCAAGAACACAGTGGAATTTGAGTCTT R: TCAGTTATATCCTGGCCACCTCTG
IL-6	F: TTCATTCCTTAGGATAGTGCTGAG R: TCCTGAGGAGTGAAGATAACAATTT
IL-8	F: AAACACACTCCACACCTTTCCAT R: GGCACACCTCATTTCCATTGAA
IL-2	F: AGTAACCTCAACTCCTGCCACAAT R: TTGCTCCATCTGTTGCTCTGTTTC
RIG-I	F: CTCCAAGAAGAAGGCTGGTTC R: AAGCAATCTATACTCCTCTAGACTTTC
LGP2	F: TCACTCCCTCCTACTCTGGCTC R: TTTCGGATCACTTCTTGCTGGTCT
IRF3	F: GGACCTGCACATTTCCAACAGC R: CAGTGACCCAGAAATCCATGTCCT
OAS	F: CCAGGGTAACTCAGGAAGGAAAGT R: CATCTCCATCAAACACGGGCTG
STA1	F: TTGACAGCAAAGTGAGAAACGTGA R: ATTGGCTTCATGTTCTCGGTTCTG
IFN-β	F: GAAATCACGCCAGTTCCAGAAG R: TCTCATTCCATCCTGTTCTAGAGATATT
IPS1	F: GACCACAAGATGTCCGCAAGC R: GGCAAGCTGTCTCTGGTGGA
TRIM25	F: TGAAACACTATATCAGGCAGTCCC R: AAATGTATGGGTTTGTGCGTGGAT
MX1	F: ATCACTGACTCGAATCCTGTACCC R: GCCTACCTTCTCCTCATATTGGCT

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
