# Peer review of "Role of CIV NS1 Protein in Innate Immunity and Viral Replication"

_ijms, 2023, doi:10.3390/ijms241210056_

Round 1

Reviewer 1 Report (Previous Reviewer 2)

1.    Figure 1A, the label in the figure is very difficult to read. The authos can increase the font size and resolution.

2.    The authirs should label the X-axis in figure 6 and Figure 8

3.    Can the author comment on the mechanism of inhibition of MDA5 by NS1?

moderate English correction is required

Author Response

Dear reviewer:

Thank you very much for your valuable comments on our manuscript. We have made relevant changes based on your suggestions. The details are as follows.

  1. Figure 1A, the label in the figure is very difficult to read. The authos can increase the font size and resolution.

According to the reviewer's request, we have increased the font size and resolution of the labels to improve the readability of the figure 1A and 1B.

  1. The authors should label the X-axis in Figure 6 and Figure 8.

According to the reviewer's request, we have labeled the X-axis in Figure 6 and Figure 8.

  1. Can the author comment on the mechanism of inhibition of MDA5 by NS1?

We investigated the regulatory effect of MDA5 gene on signal pathway in dogs, and found that MDA5 mainly activates IFN-β, IRF3 and NF-κB promoters through CARD region to conduct signals downstreamFu, C.; Ye, S.; Liu, Y.; Li, S. Role of CARD Region of MDA5 Gene in Canine Influenza Virus Infection. Viruses 2020, 12, doi:10.3390/v12030307.)。In this study, we found that NS1 protein inhibited the activation of IFN and IRF3 by the MDA5 CARD domain, but did not affect the interaction of MDA5 with other proteins of influenza virus

  1. Moderate English correction is required.

At the request of the reviewers, we have had the manuscript extensively revised for grammar by native English-speaking professionals.

Reviewer 2 Report (New Reviewer)

In the study entitled "Role of CIV NS1 Protein in Innate Immunity and Viral Replication", Fu et al. investigated the role of the influenza viral protein NS1 in the regulation of the innate immunity and demonstrated that this protein antagonizes MDA5 function. This work is relevant in the context of viral infection due to the abiliy of this virus to infect both dogs and humans. I have nonetheless some concerns that the authors should address before publication.

1) Several results have been obtained from artificial tools such as luciferase activity or overexpression experiments. Especially, for the interaction studdy where viral proteins and MDA5 are overexpressed, the levels of expression of exogenous proteins can highlight interactions that do not exist naturally. If possible, I strongly recommend repeating these experiments in a natural context with addressing the interaction between viral proteins (produced after infection) with endogenous MDA5. Similarly, the authors investigated the effect of viral proteins on the ability of MDA5 to activate several cytokine promoters such as IFNb, IRF3. I would recommend checking the levels of endogenous cytokines mRNAs and proteins.

2) Figure 1a: A negative control is missing with a protein known to not interact with MDA5 for example.

3) Lane 139-143: "Studies have shown that NS1 can selectively increase the synthesis of viral mRNA, resulting in the preferential synthesis of viral proteins. To test whether the bifc signal was the result of NS1 stimulating RNP subunit synthesis, western blot analysis was performed on cells transfected with the plasmid encoding RNP subunit in the presence and absence of NS1, respectively(Figure 2 below).":  To my opinion, this point cannot be addressed with the design of the experiment as the viral mRNA are not under the control of their native promoters. If the authors want to address this question, I would recommend to overexpress NS1 under infection condition.

4) All the Western blot should be quantified.

5) Figure 3: What is the effect of the viral proteins when only polyIC or MDA5 are present? The authors should show a Western blot for all the conditions.

Minor comments:

Lane 17: MDA5 interacts with the viral proteins, not with the plasmids

Figure 4c: Replace lucigerase by luciferase

Lane 230: The authors have to precise that cells were transfected with the IFNb or IRF3 luciferase plasmids. 

Author Response

Dear reviewer:

Thank you very much for your valuable suggestions on our manuscript. We have made relevant changes based on your suggestions. The details are as follows.

In the study entitled "Role of CIV NS1 Protein in Innate Immunity and Viral Replication", Fu et al. investigated the role of the influenza viral protein NS1 in the regulation of the innate immunity and demonstrated that this protein antagonizes MDA5 function. This work is relevant in the context of viral infection due to the abiliy of this virus to infect both dogs and humans. I have nonetheless some concerns that the authors should address before publication.

1) Several results have been obtained from artificial tools such as luciferase activity or overexpression experiments. Especially, for the interaction studdy where viral proteins and MDA5 are overexpressed, the levels of expression of exogenous proteins can highlight interactions that do not exist naturally. If possible, I strongly recommend repeating these experiments in a natural context with addressing the interaction between viral proteins (produced after infection) with endogenous MDA5. Similarly, the authors investigated the effect of viral proteins on the ability of MDA5 to activate several cytokine promoters such as IFNb, IRF3. I would recommend checking the levels of endogenous cytokines mRNAs and proteins.

Thank you very much for your valuable suggestions, to verify that CIV proteins and NS3 interact in vivo, we used mouse anti-NP/NS1 antibody and rabbit anti-MDA5 antibody in CIV-infected or non-infected 293T cells, confocal images showed that NP, NS1 and MDA5 colocalized in the cytoplasm (Figure 1C). In addition, we examined the effect of MDA5-CARD on IFNβ and IRF3 mRNA expression in cells transfected with Flag-CMV or Flag-NS1, and found that the NS1 protein of H3N2 significantly antagonized the promotion of the IFNβ and IRF3 mRNA by MYC-MDA5-CARD (P <0.05) (Figure 4E).

2) Figure 1a: A negative control is missing with a protein known to not interact with MDA5 for example.

Thank you very much for pointing out our mistake. We have added negative control to Figure1A and 1B, by using MYC-CMV that known to not interact with MDA5.

3) Lane 139-143: "Studies have shown that NS1 can selectively increase the synthesis of viral mRNA, resulting in the preferential synthesis of viral proteins. To test whether the bifc signal was the result of NS1 stimulating RNP subunit synthesis, western blot analysis was performed on cells transfected with the plasmid encoding RNP subunit in the presence and absence of NS1, respectively (Figure 2 below).":  To my opinion, this point cannot be addressed with the design of the experiment as the viral mRNA are not under the control of their native promoters. If the authors want to address this question, I would recommend to overexpress NS1 under infection condition.

Thank you very much for your comments, we transfected MYC-NS1 or MYC-CMV plasmid into 293T cells infected with CIV virus, and detected the relative expression of PA, PB1, PB2 and NP mRNA, and found that overexpression of NS1 had no significant effect on the expression of the above gene mRNA (Figure 2B).

4) All the Western blot should be quantified.

According to the reviewer's request, we have quantified all the Western blot in Figure 2A, Figure3, Figure4D and Figure5D.

5) Figure 3: What is the effect of the viral proteins when only polyIC or MDA5 are present? The authors should show a Western blot for all the conditions.

Thank you very much for your comments. In Figure3, we added the activation of IFN-β promoter when only MDA5 or POLY IC existed, and found that the addition of MDA5 and POLY at the same time had the strongest effect on the activation of IFN-β promoter. Therefore, the co-existence of MDA5 and POLYIC was selected to detect the effects of CIV proteins on IFN-β. Meanwhile, the expressions of MDA5 and CIV proteins were detected by western blot, and MDA5 protein was quantitatively analyzed.

Minor comments:

Lane 17: MDA5 interacts with the viral proteins, not with the plasmids

Thank you very much for your comments, we have changed the description into MDA5 interacts with the viral proteins.

Figure 4c: Replace lucigerase by luciferase

Thank you very much for your comments, we have replaced lucigerase by luciferase.

Lane 230: The authors have to precise that cells were transfected with the IFNb or IRF3 luciferase plasmids. 

Thank you very much for your comments, we have shown at lane230 that cells were transfected with the IFNb or IRF3 luciferase plasmids.

This manuscript is a resubmission of an earlier submission. The following is a list of the peer review reports and author responses from that submission.

Round 1

Reviewer 1 Report

The manuscript by Cheng Fu et al needs to improve the representation of figures and legends. None of the data presented in the article has a legend and it won't be easy for readers to follow based on the text (result section). Figures are not represented in a proper format.  Moreover, this work does not provide any novelty in the community. NS1 and MDA5 have been already shown to interact with each other. Also, NS1 NS1 has been shown to antagonise the production of IFN-β by interacting with and promoting the degradation of RIG-I and  MDA5. 

Author Response

Dear reviewer:

Thank you very much for your valuable comments on our manuscript. We have made relevant changes based on your suggestions. The details are in the word file.

Reviewer 2 Report

1.     The authors should describe the figure legends properly. They should mention in the legend what is present in each panel of every figure. Otherwise, it is really difficult to interpret the result

2.     They should mention the number of replicates in each experiment and how the error bars were calculated

3.     There are several grammatical mistakes

4.     Did the author check if the interaction between MDA5 and NS1 is mediated by nucleic acid or not? They can simply comment on this by adding nuclease in the lysis buffer during Co-IP

5.     In the discussion section, the author should discuss the innate immune evasion strategy of different RNA and DNA viruses., There are few recent articles that highlight different strategies of DNA and RNA viruses to antagonize the innate immune response (1. M Bhattacharya, D Bhowmik, Y Tian, H He, F Zhu, Q Yin, Journal of Biological Chemistry, 102986; 2. D Bhowmik, M Du, Y Tian, S Ma, J Wu, Z Chen, Q Yin, F Zhu, Nucleic acids research 49 (16), 9389-9403)

6.     Bimolecular fluorescence complementation results are difficult to interpret. The authors should use high resolution data and interpret results properly.

Author Response

(The authors gave the same response as above.)

Round 2

Reviewer 1 Report

NA

Reviewer 2 Report

I have no further comments